# Enhanced Recovery of Zn from Carbonate-Type Mixed Oxidized Ore (CMO) by Combining Organic Acid Leaching with Mechanical Activation

**Hao Deng [1], Xuelin Pan [1], Fanyun Chen [1], Qingshan Gao [1], Chen Tian [1,2,*] and Zhang Lin [1,2]**

[1] School of Metallurgy and Environment, Central South University, Changsha 410083, China
[2] Chinese National Engineering Research Center for Control & Treatment of Heavy Metal Pollution, Changsha 410083, China
* Correspondence: birdytc@hotmail.com

**Abstract:** With excessive consumption of sulfide ores, using low-grade Pb-Zn mixed ores to obtain Zn has attracted more attention. Acid leaching is an effective method for Zn recovery; however, a high concentration of inorganic acid brings severe environmental issues, including acidic wastewater, contaminated soil, etc. Compared with inorganic acid, organic acid showed lower acidity and toxicity. Herein, we applied an effective method for Zn recovery from carbonate-type Pb-Zn mixed ore (CMO), by combining organic acid leaching with mechanical activation. Among the organic acids applied (lactic, malonic, citric, amber, acetic, and tartaric one), lactic acid was selected for its high leaching efficiency. The optimum condition was identified via response surface methodology, with a lactic acid concentration of 1.15 mol/L, L/S ratio of 20, leaching time of 75 min, and temperature of 75 °C. The effect of mechanical activation on Zn leaching was further investigated. The leaching efficiency increased to 90.1% after mechanical activation, which was the highest leaching efficiency for organic acid leaching. Mineralogical characterization showed that the bulk minerals were milled into small particles while the proportion of amorphization increased. Such activation effects improved the acid-solubility of Zn speciation. This work provided a potential green method for metal recovery from low-grade sources.

**Keywords:** organic acid leaching; Zn recovery; ball milling; low-grade ore

## 1. Introduction

Zinc (Zn) is one of the most commonly used non-ferrous metals in the global industry. The global Zn smelter production was estimated to be 13.8 Mt in 2020 [1]. Over 86% of Zn comes from primary sources (ores and concentrates). For primary sources, Zn sulfide ores account for 95% and non-sulfide resources account for 5% [2]. Carbonate-type mixed oxidized ore (CMO) is the main type of non-sulfide source, which comes from the weathering effect of sulfide ores. With increasing consumption of Zn, the reserve of Zn sulfide ores is decreasing. Therefore, it is essential to develop a technique for recovering Zn from low-grade non-sulfide resources.

Presently, non-sulfide Zn ore is considered as one of the alternative resources of Zn. However, non-sulfide Zn ores contain comparable levels of impurities (e.g., silica, gypsum, calcite, hematite), resulting in low-grade Zn concentration [1,3]. The hydrometallurgy process is the main method of Zn recovery from sulfide resources, of which 80% of Zn is recovered using inorganic acid (e.g., sulfuric acid and chloride acid). However, inorganic acidic leaching is not suitable for non-sulfide oxide ores. Firstly, hydrochloric acid should work at pH < 1 to obtain considerable leaching efficiency. Lupi [4] found that the recovery yield of Zn is 96.5% in 1.5 mol/L $H_2SO_4$ for ZnO leaching, while only 73.9% of Zn was recovered in 0.5 mol/L $H_2SO_4$. Such highly acidic leaching solution leads to severe environmental issues and corrosion of the equipment. Secondly, inorganic acids dissolve all metal

elements with poor selectivity, leading to the co-leaching of Fe in Zn-bearing ores, which causes an additional purification process in smelter production. Alkaline/ammoniacal leaching is effective for removing the most undesired elements (e.g., Fe, Ca, Mg, Cd) at a high pH value [1]. The drawback of alkaline leaching is the requirement for high reagent consumption, as well as a lower recovery rate than acid leaching. Liu [5] used 6 mol/L NaOH to recover Zn from lean-oxidized ores with a maximum Zn dissolution of 85.5%. Alkaline leaching also produces residues containing multi-heavy metals, which are regarded as hazardous wastes [6,7].

Therefore, regarding environmental concerns, low-molecular-weight organic acids (LMWOAs) are used as emerging leaching reagents for hydrometallurgy processes [8]. LMWOAs are biologically degradable. Furthermore, organic acid leaching is not restricted to low pH conditions, indicating a mild condition. Halli et al. [9] used citric acid to recover Zn from the electric arc furnace (EAF) dust at pH = 3.7. However, the leaching efficiency is not comparable to inorganic acid without pretreatment [10]. Mechanical activation was reported to be an effective method for enhancing the reactivity of minerals. Mechanical activation can change the phase composition, micro-structures, and degree of crystallinity. A low degree of crystallinity was reported to accelerate the reaction reactivity [11,12]. Parviz et al. [13] enhanced the Zn leaching efficiency of Zn plant residues from 82.4% to 99.9% via ball milling pretreatment. The applied mechanical energy increased the proportion of the amorphous region in the structure, leading to the leaching reactivity enhancement of ZnO particles. Therefore, mechanical activation is expected to reduce the reagent consumption and elevate the leaching efficiency. However, the coupling effect of mechanical activation and organic acid leaching on metal recovery has rarely been investigated.

In this study, we explored an enhanced Zn recovery from CMO through the combined treatment of organic acid leaching and mechanical activation. The leaching performance of the common six LMOWAs was tested (lactic, malonic, citric, amber, acetic, and tartaric one) and lactic acid was chosen as the best organic acid. Then, the coupling effect of ball-milling pretreatment with organic leaching was investigated. Results showed that the leaching efficiency increased from 44.5% to 90.1%. It was found that enhancement of leaching reactivity was related to the increased exposure of small sphalerite particles, as well as the increasing degree of amorphization by ball-milling. This work confirmed the coupling effect of ball milling and organic acidic leaching for Zn recovery from CMO. This combined technique would provide a potential green method for metal recovery from low-grade sources.

## 2. Materials and Methods

The carbonate-type mixed oxidized ore (CMO) was obtained from Shaoguan Pb/Zn smelter, Guangdong Province of Southern China. The CMO sample was collected and kept in vacuum drying overnight. Then, the CMO sample was ground and sieved through a 300 mesh grid to obtain powders with a size smaller than 50 μm.

The inorganic acids (nitric acid, sulfuric acid, and hydrochloric acid) were all guaranteed reagents (GR) and purchased from Sinopharm Chemical Reagent Co., Ltd. in Beijing, China. The organic acids (lactic acid, malonic acid, citric acid, amber acid, acetic acid, and tartaric acid) were all analytically pure (AR). Lactic acid, citric acid, tartaric acid, and acetic acid were all and obtained from Sinopharm Chemical Reagent Co., Ltd. in Beijing, China. Malonic acid and amber acid were purchased from Macklin.

### 2.1. Mineralogical Characterization

The chemical composition of CMO was analyzed by X-ray fluorescence (XRF, XRF-1800, Shimadzu, Japan). Mineralogy was investigated by X-ray diffraction (XRD, Bruker D8 ADVANCE, Bruker, Germany) with Cu K$\alpha$ radiation ($\lambda$ = 1.54060 Å), and the test was conducted at 40 kV and 40 mA. Diffraction data recorded ranged from 10° to 70° 2-Theta with a scan rate of 1.2 o/min. The phase compositions were identified using

Crystallography Open Database (COD). The phase quantification was analyzed based on the Rietveld method. The compositional variation of CMO was analyzed by attenuated total reflectance infrared (ATR−IR) spectroscopy (ATR-IR, Nicolet iS50, ThermoFischer Scientific, Madison, WI, USA) equipped with a diamond accessory.

*2.2. Experimental Procedures*

2.2.1. Leaching Methods

The leaching experiment was performed in a 50 mL conical flask. The flask was placed in a water bath. A digital magnetic stirrer apparatus was utilized for heating and equipped with a digital temperature controller. In a typical leaching process, 1 g of CMO powder and a stirrer were added into the flask, followed by 10 mL of organic acid solution (1 M). The flask was sealed and transferred into a water bath with the target temperature modulated previously. The reaction time was set as one hour initially. Timing started when the stirring switched to a stirring speed of 250 rpm. The whole reaction time was set. All chemicals used were analytical-grade purity.

To determine the best performance, the common LMWOAs were chosen as leaching agents according to several summarized articles. The reaction condition was based on the typical leaching process described above.

2.2.2. Analysis Method

Inductively Coupled Plasma Optical Emission Spectrometry (ICP-OES, Avio500, PerkinElmer, Singapore) was used for determining the concentration of zinc in the leaching solutions. Before the test, the obtained leaching solution was filtered through a membrane filter with a 0.22 μm pore size. The residue was washed with ultra-pure water three times. Then, the solid was dried in a vacuum oven overnight. XRD, SEM-EDS, and ATR-IR were employed for characterizing the residue.

2.2.3. Experimental Design for Leaching Condition Optimization

In the leaching experiment, at least 6 reaction parameters should be optimized: stirring speed, pH, temperature, time, liquid-to-solid ratio (L/S), and organic concentration. The stirring speed was not taken into consideration because it has been reported to be less effective in improving leaching efficiency [14]. Thus, the stirring speed was fixed at 250 rpm.

In order to identify the reproductivity of results, both leaching experiments were carried out three times. The mean values and deviations were further calculated and displayed in the figures.

The pH determined the chemical speciation of the organic acid, and we investigated the influence of pH solely. The optimum pH value for organic acid leaching was fixed at pH = 2. A further mechanism is discussed in the later section.

It was demonstrated that four parameters should be optimized, and the interaction among factors was analyzed via response surface methodology analysis. Four parameters at five levels of experimental design (Table 1) were conducted [13,15]. The central composite design (CCD) was applied in investigating the optimum leaching condition. In general, the following pertinent factors should be considered in optimization, namely, pH, temperature, time, solid-to-liquid ratio, organic acid concentration, and stirring speed. The parameter range for the experiment was based on the summary of previous references (Table S1). For that, 30 experimental runs including 6 central points were achieved by the CCD prearranged in Table S2.

**Table 1.** Element composition of CMO.

| Element | Zn | Pb | Fe | S | Al | Ca | Mg | Others |
|---|---|---|---|---|---|---|---|---|
| Content (wt.%) | 28.6 | 14.9 | 10.5 | 16.9 | 0.542 | 3.10 | 0.512 | 25.5 |

### 2.2.4. Mechanical Activation Treatment

In a typical-ball milling procedure, 10 g of CMO powders were mixed with agate balls at a ball-to-mass ratio of 1/1 (B/M) and transferred into an agate pot with a volume of 250 mL. A pair of agate pots was then fixed in a laboratory-scaled planetary ball mill (YXQM-4L, MITR, Changsha, China). The milling period was set at 30 min along with a rest for 5 min and rotational speed was maintained at 500 rpm under ambient conditions.

## 3. Results and Discussion

### 3.1. Characteristics of CMO

The element composition of CMO obtained from the XRF analysis is listed in Table 1. It can be seen that Zn was the main element in CMO (28.6 wt.%), while Pb and Fe were associated elements with contents of 14.9 wt.% and 10.5 wt.%, respectively. It was revealed that the grade of Zn in CMO (28.6 wt.%) was lower than that in sulfide ore (~65 wt.%) [16].

The phase composition was then analyzed through XRD (Figure 1A). It can be seen that the Zn-bearing phases include sphalerite (ZnS) of 21.9 wt.%, smithsonite ($ZnCO_3$) of 14.1 wt.%, and willemite ($Zn_2SiO_4$) of 1.86 wt.%. According to XRF and XRD results, the proportions of Zn in Zn-bearing phases were 63.5% (sulfide), 4.7% (silicate), and 31.8% (carbonate). Therefore, the main Zn speciation in CMO was acidic soluble. Other mineral phases without Zn were suggested to be siderite ($FeCO_3$) of 20.0 wt.%, gypsum ($CaSO_4\cdot2H_2O$) of 13.1 wt.%, anglesite ($PbSO_4$) of 4.71 wt.%, cerussite ($PbCO_3$) of 8.09 wt.%, quartz ($SiO_2$) of 7.35 wt.%, pyrite ($FeS_2$) of 5.61 wt.%, and calcite of ($CaCO_3$) 3.21 wt.%.

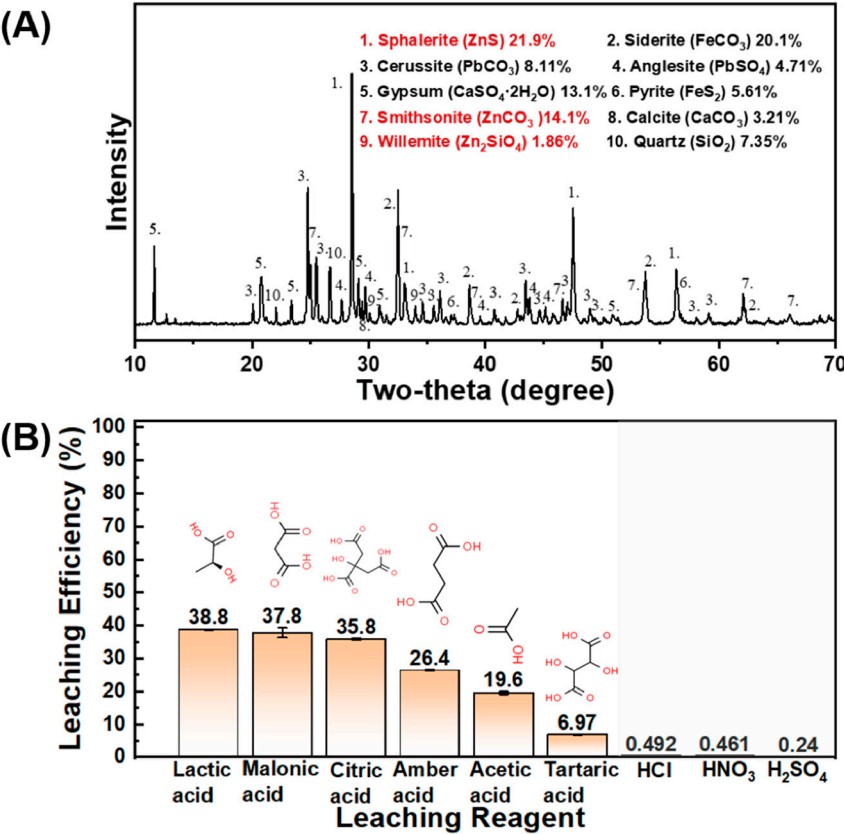

**Figure 1.** (**A**): XRD pattern of original CMO; (**B**): leaching efficiency of organic acids [acid concentration: 1 (M), liquid/solid ratio: 10 mL: 1 g; time: 1 h; temperature: 60 °C; pH: 2].

### 3.2. Zn Leaching Effects of Organic Acids

Typical mono-, di-, and tricarboxylic acids were utilized to investigate the Zn leaching effect of CMO. As shown in Figure 1B, among tested organic acids, lactic acid showed the highest leaching efficiency of 38.8%, while malonic acid showed comparable leaching

efficiency of 37.8%. For the rest of the organic acids, the leaching effect showed an order of citric acid (35.8%) > amber acid (26.4%) > acetic acid (19.6%) > tartaric acid (6.97%). Although tartaric acid showed the lowest leaching efficiency among six tested organic acids (6.97%), it was much higher than that of the inorganic acid with a leaching efficiency below 0.5%. According to previous reports, acidolysis was regarded as the main effect of acid leaching [17]. For the acidolysis process, pH value played an important role in both inorganic and organic acid leaching. Lower pH indicated a higher concentration of $H^+$, which was significant for Zn dissolution. It can be seen that inorganic acids ($H_2SO_4$, HCl, and $HNO_3$) showed almost no effect on Zn leaching with 0.5% Zn dissolution at pH = 2 (Figure 1B) or higher value (Figure S1). This phenomenon implied that inorganic acid could not provide enough $H^+$ to dissolve Zn from CMO. Compared with inorganic acid, organic acid reserved protons in organic functionalized groups [18]. Thus, organic acid could provide extra $H^+$ to dissolve metal even at pH = 2 (Figure 1B). As shown in Table S7, the pH values of the lixiviums were over 6 by inorganic acid leaching, while the pH values of the lixiviums were kept around 4~5 by organic acid leaching. It was a 10~100-fold difference in concentration of $H^+$ between organic acid leaching and inorganic acid leaching. The lower pH values achieved by organic acid leaching indicated that organic acids released more $H^+$ than inorganic acids for acidolysis. Thus, organic acid leaching showed a much higher leaching efficiency than that of inorganic acid.

The Zn leaching efficiency was highly relative to the properties of different organic acids, with an order of lactic acid (38.8%) > malonic acid (37.7%) > citric acid (35.8%) > amber acid (26.4%) > acetic acid (19.5%) (Figure 1B). Moreover, the p$Ka$ values of these organic acids showed an order of the following: malonic acid (2.43) < tartaric acid (2.72) < citric acid (3.06) < amber acid (3.55) < lactic acid (3.78) < acetic acid (4.54) (Table 2). The p$Ka$ value reflected the acidity intensity of organic acid in the solution, of which smaller p$Ka$ values revealed stronger acidity and capability of acidolysis [8]. Therefore, it is worth noting that the order of the Zn leaching effect was consistent with the order of acidity intensity for partial organic acids. However, some controversies remained, one of which was that tartaric acid has the second strongest acidity intensity (2.72), whereas tartaric acid showed the lowest Zn leaching efficiency (6.96%). This might be due to the extremely low solubility of zinc tartrate in water (Ksp = 0.022 g/100 mL at 20 °C) [19], which greatly reduced the Zn leaching efficiency in solution. Another controversy was that lactic acid had moderated acidity, while the Zn leaching efficiency by lactic acid was the highest among six organic acids (38.8%). This could be explained in view of metal selectivity during multi-metal leaching. Among the selected six organic acids, lactic acid had the highest selectivity of Zn and the lowest selectivity for Fe and Pb (Figure S2). The selectivity could also be predicted through the stability constants (lg$K$) of the metal–organic complex (Table 2). Except for lactic acid, the lg$K$(Pb) values for other organic acids (6.50 for citric acid, 2.80 for amber acid, 2.52 for acetic acid, and 3.78 for tartaric acid) were much larger than lg$K$(Zn) values (4.71 for citric acid, 1.60 for amber acid, 1.50 for acetic acid, and 2.68 for tartaric acid). In the case of lactic acid, lg$K$(Pb) (2.40) was close to lg$K$(Zn) (2.20). Therefore, compared with other organic acids, lactic acid had a relatively stronger selectivity for Zn with a poorer selectivity for Pb. In addition to the stability constant of the complex, the steric factor was also important for sufficient Zn leaching by organic acid. The steric factor could be estimated by the McGowan characteristic molecular volume [15,20,21]. As shown in Table S8, the order of A values is listed as follows: acetic acid (0.50) < lactic acid (0.67) < malonic acid (0.70) < amber acid (1.04) < tartaric acid (1.05) < citric acid (1.30). By comparison, lactic acid had the second smallest molecular volume among the six selected organic acids, leading to a more effective mass diffusion [22]. The lactate complex $Zn[(C_3H_5O_3)_2(H_2O)_2] \cdot H_2O$ has also been proven to be highly soluble in water [23]. It could be concluded that lactic acid had moderated acidity and small steric hindrance, which led to higher selectivity for Zn than Pb. Based on the comprehensive analysis, lactic acid had the best performance on Zn leaching among the selected organic acids.

**Table 2.** Physicochemical properties of organic acids and corresponding metal complex.

| | $pK_a$ [1] | $lgK_1(Zn)$ [2] | Solubility of Zn-Complex [2] | $lgK_1(Fe)$ [2] | $lgK_1(Pb)$ [2] |
|---|---|---|---|---|---|
| lactic acid | 3.78 | 2.20 | Soluble | 7.10(3+) | 2.40 |
| malonic acid | 2.43 | - | Soluble | - | - |
| citric acid | 3.06 | 4.71($HL^{2-}$) | Slightly soluble | 3.08(2+) 12.5(3+) | 6.50 |
| amber acid (succinic acid) | 3.55 | 1.60 | Soluble | 7.49(3+) | 2.80 |
| acetic acid | 4.54 | 1.50 | 30.0 g/100g saturated solution | 3.20(2+) | 2.52 |
| tartaric acid | 2.72 | 2.68 | 0.0220 g/100 mL | 7.49(3+) | 3.78 |

[1] Data collected from http://www.chemicalize.com. [2] Data collected from Lange's Handbook of Chemistry, Sixteenth Edition [24].

### 3.3. Leaching Condition Optimization

#### 3.3.1. pH Optimization

The initial pH value influences both the concentration of $H^+$ and the speciation distribution of lactic acid, which would further affect the leaching efficiency. Therefore, the influence of pH was investigated first. The corresponding effects were shown in Figure S3 in a wide range of pH values (e.g., pH = 1, 3, 4, 6), the leaching efficiency of lactic acid was maintained at approximately 10%. The Zn leaching efficiency of 10% was close to that of inorganic acid leaching at pH = 1 (Figure S1). This phenomenon showed a proton buffer effect during the metal leaching process, in which the proton concentration was maintained at a stable level [25,26]. It was proposed that the proton buffer effect of the weak acid could maintain the pH value in the solution. Thus, the Zn leaching efficiency was maintained at approximately 10%.

It is worth noting that the Zn leaching efficiency increased to 38.8% when the pH value was close to 2 (Figure S2). It has been reported that organic acids had a neutral charge in solution at pH = 2 [27]. Thus, the charge of lactic acid changed from a negative state to a neutral state when the pH decreased from 6 to 2. Similarly, the charge of lactic acid changed from a positive state to a neutral state when pH increased from 1 to 2. Owing to the neutral charge of lactic acid, the effect of chelation/complexolysis between lactic acid and $Zn^{2+}$ was maximized [20]. Therefore, the highest Zn leaching efficiency was obtained at pH = 2, in the pH range of (1–6). Although lactic acid leaching showed good performance at lower pH (<1), it brought environmental pollution and corrosion of equipment. For the purpose of green production, Zn leaching should be operated in mild conditions. In the end, the optimized pH was fixed at 2.

#### 3.3.1. RSM Model Fitting, Evaluation, and Confirmation

RSM optimization on 4 factors (temperature, time, organic acid concentration, and L/S ratio) was applied to optimize leaching conditions except for pH. Four kinds of models were tested for fitting the model, including the linear model, 2FI model, quadratic model, and cubic model. Four factors of sequential *p*-value, lack of fit *p*-value, adjusted R2, and predicted R2 were utilized to briefly evaluate these models. The fitting summary is shown in Table S4. Compared with the other three models, the quadratic model has the smallest sequential *p*-value (0.0166), the smallest lack of fit *p*-value (0.786), the largest adjusted R2 (0.939), and the largest predicated R2 (0.882). Thus, it was suggested that a quadratic model be applied for interaction analysis and optimization [28].

Four single factors (time, denoted as A; lactic acid concentration, denoted as B; temperature, denoted as C; and ratio of liquid/solid, denoted as D) were further analyzed by analysis of variance (ANOVA) of the quadratic model. Furthermore, the interaction factors (AB, AC, AD, BC, BD, and CD), as well as self-quadratic factors (A2, B2, C2, and D2) were also analyzed. The results were displayed in Table S4. In the ANOVA, the *p*-value was used

to determine whether a model was significant. If the *p*-value of the model was smaller than 0.0500, it was demonstrated that the model was significant [29]. In this study, the *p*-value of the quadratic model was less than 0.0166, revealing the model's significance. It is also worth noting that the *p*-values of A, B, C, D, AC, BD, B2, and C2 were all less than 0.0500, implying that these factors were significant model parameters. These results suggested that the above four single factors could influence Zn leaching efficiency individually. Moreover, the AC factor indicated that leaching efficiency was affected by the interaction between time and temperature, while the BD factor implied that leaching efficiency was influenced by the interaction between lactic acid concentration and the ratio of liquid/solid. However, the *p*-values of AB, AD, CD, and A2 were 0.966, 0.752, 0.813, and 0.829, respectively, suggesting that these interaction factors were insignificant ($p > 0.5$) for Zn leaching. It was necessary to delete these factors to further revise the model. After model revision, the *p*-values of all factors were less than 0.5 (Table S6), revealing that the revised quadratic regression model was sufficient for Zn leaching. The final equation in terms of actual factors was described as follows:

$$\text{Leaching Efficiency} = -83.896 + 0.497 \times A + 61.926 \times B + 1.164 \times C + 1.687 \times D - 0.006 \times AC - 0.696 \times BD - 20.833 \times B^2 - 0.003 \times C^2 \tag{1}$$

The diagnostic plots in Figure S4a showed an approximate linearity normality of the data, indicating a good fitting in the model. The diagnostic plots in Figure S4b suggest that the experimental results were in good agreement with predicted values ($R2 = 0.968$).

### 3.3.2. Interactions among Factors and Optimization

The interactive effects of independent factors on Zn leaching efficiency were investigated via 3D response surface plots (Figure S5) and corresponding projection contour maps (Figure 2).

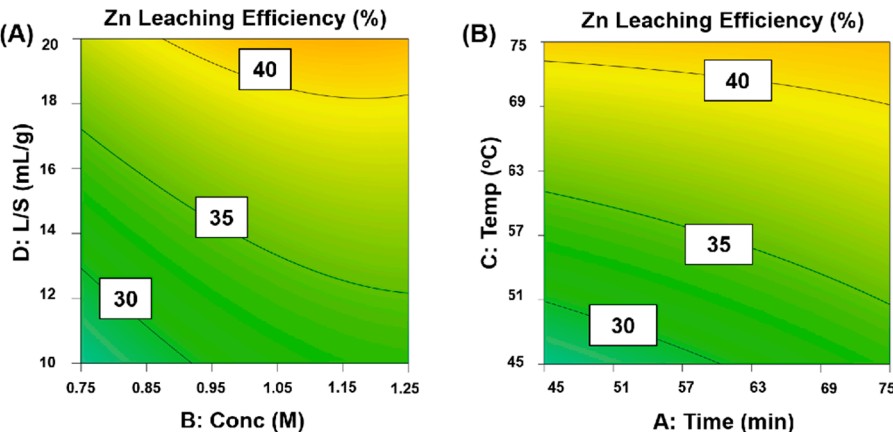

**Figure 2.** The contour maps for Zn leaching efficiency vs. (**A**) L/S ratio and lactic concentration [reaction time: 60 min, temperature: 60 °C]; (**B**) temperature and time [L/S ratio: 15 mL/g, lactic acid concentration: 1 mol/L].

The interactions between lactic acid concentration and the ratio of liquid/solid were investigated by fixing the other two factors at a central level. The results were shown in a 3D surface plot (Figure S5a) and a contour map (Figure 2A). As shown in Figure S5a, the Zn leaching efficiency increased almost linearly as the L/S increased from 10 to 20 mL/g. The Zn leaching efficiency also increased with the increase in lactic acid concentration from 0.75 to 1.25 mol/L. However, the lactic acid concentration had little effect on the increase in Zn leaching efficiency at a higher ratio of L/S (Figure 2A). This phenomenon was probably due to the accumulation of lactic acid/lactate molecules on the surface of CMO. The coverage of organic molecules hindered the mass transformation from the surface of the CMO solid

to the bulk solution [6,30]. The results showed that a higher Zn leaching efficiency could be obtained at a higher ratio of L/S and a moderate lactic acid concentration.

The interactive effect of time and temperature on Zn leaching efficiency was investigated by fixing the ratio of L/S and lactic acid concentration. The results were shown in Figures S4b and 2B. The contour plot in Figure 2B illustrated that Zn leaching efficiency increased almost linearly with the increase of leaching time from 45 min to 75 min. Zn leaching efficiency also showed a nearly linear increase as the temperature increases from 45 °C to 75 °C. It is obvious that the contour of temperature was denser than that of leaching time, indicating that temperature had a more significant effect on the increase in Zn leaching efficiency than leaching time. The results showed that the longer leaching time and higher temperature were suitable for a higher Zn leaching efficiency.

### 3.3.3. Optimization and Comparison of Zn Leaching between the Predicted and Actual Value at Optimized Condition

The optimization criteria were the highest Zn leaching efficiency. The optimized conditions for Zn leaching were kept within the studied range. As presented in Table S9, software generated 20 solutions with high Zn leaching efficiency. Owing to its highest predicted Zn leaching efficiency (46.5%), solution No.1 was chosen as the optimized condition. The optimized condition was as follows: time of 75 min, temperature of 75 °C, L/S ratio of 20, and lactic acid concentration of 1.15 mol/L. Under the optimized condition (Table 3), the experimentally obtained value (44.5%) was close to the predicted value (46.5%). Regarding the fact that the model-predicted value was within the 90% confidence interval, the constructed quadratic model was suitable for predictive purposes [13,31].

**Table 3.** Comparison of Zn leaching efficiency between predicted and experimental values at the optimized condition.

| pH | Lactic Acid Concentration | Temperature | Time | L/S | Zn Leaching Efficiency (%) | |
|---|---|---|---|---|---|---|
| 1 | (mol/L) | (°C) | (min) | | Predicted | Experimental |
| 2 | 1.15 | 75 | 75 | 20 | 46.5 | 44.5 |

### 3.4. Combined Effect of Mechanical Activation on Zn Leaching

It can be seen from the experimental result (Table 3) that the optimized Zn leaching efficiency was only 44.54% via lactic acid leaching, which was far from satisfactory. According to the composition analysis of leaching residue (Figure S6), it is proposed that the Zn-bearing carbonate was totally dissolved (31.8% of total Zn). However, sphalerite was conserved in the CMO (55.5% of total Zn). To enhance the Zn leaching efficiency, a combined method of mechanical activation (ball-milling treatment) and organic acid leaching was applied for Zn recovery. The effects of three crucial parameters in ball-milling treatment (rotation speed, ball/mass ratio, and rotation time) on Zn recovery were investigated.

### 3.4.1. Effect of Rotation Speed

The effect of rotation speed on Zn leaching efficiency is shown in Figure 3A. It can be seen that the Zn leaching efficiency increased from 44.5% to 51.9% when the rotation speed increased to 500 rpm. XRD results showed no changes in the composition of CMO after ball-milling treatment (Figure 3B). However, the BET surface area increased from 3.28 $m^2$/g to 5.82 $m^2$/g after treatment with 500 rpm for 1 h (Table S10). Therefore, the enhancement of Zn leaching efficiency was probably attributed to the increase in specific area of CMO.

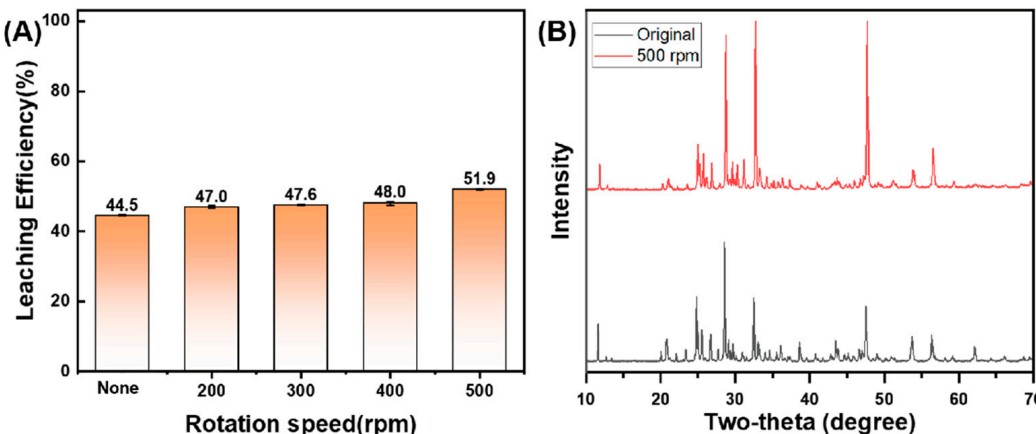

**Figure 3.** (**A**): Effect of rotation speed on leaching efficiency [milling time: 1 h, ball-mass ratio: 1:1]; (**B**) XRD pattern of original CMO and CMO treated at 500 rpm.

3.4.2. Effect of Ball-Mass Ratio

The effect of ball/mass ratio on Zn leaching efficiency was investigated. The results are displayed in Figure 4A. The Zn leaching efficiency increased from 51.9% to 73.2% when the ball/mass ratio increased from 1:1 to 40:1. This phenomenon indicated that the leaching reactivity was elevated at a higher ball/mass ratio. Such enhancement was supposed to be related to the change of composition in CMO. The XRD pattern of treated CMO showed the broadening of the diffraction peaks and decreased intensity of diffraction peaks (Figure 5B). This observation indicated the existence of the amorphous region [32]. As shown in Figure S7, the degree of amorphization increased from 12.2% to 47.1% when the ball/mass ratio increased from 1:1 to 40:1. This result suggests that crystallite phases (e.g., ZnS, ZnCO$_3$) were transformed to the amorphous phases by mechanical activation.

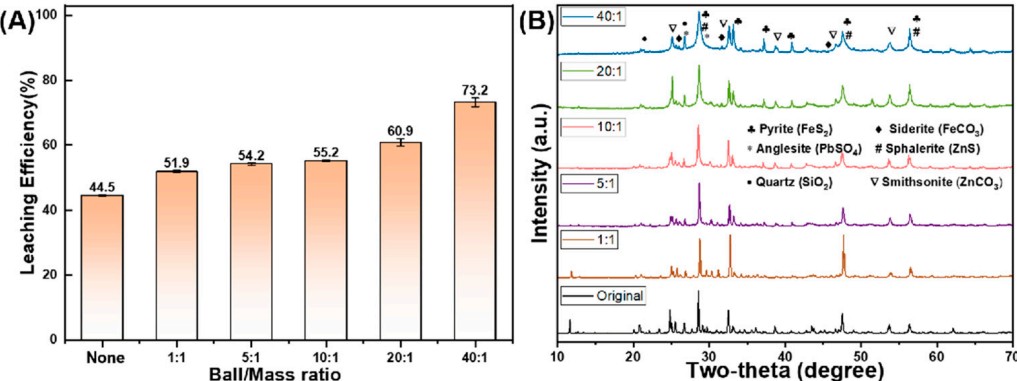

**Figure 4.** (**A**): Effect of ball/mass ratio on leaching efficiency: [time: 1 h, rotation speed: 500 rpm]; (**B**): XRD pattern of CMO under different ball/mass ratio.

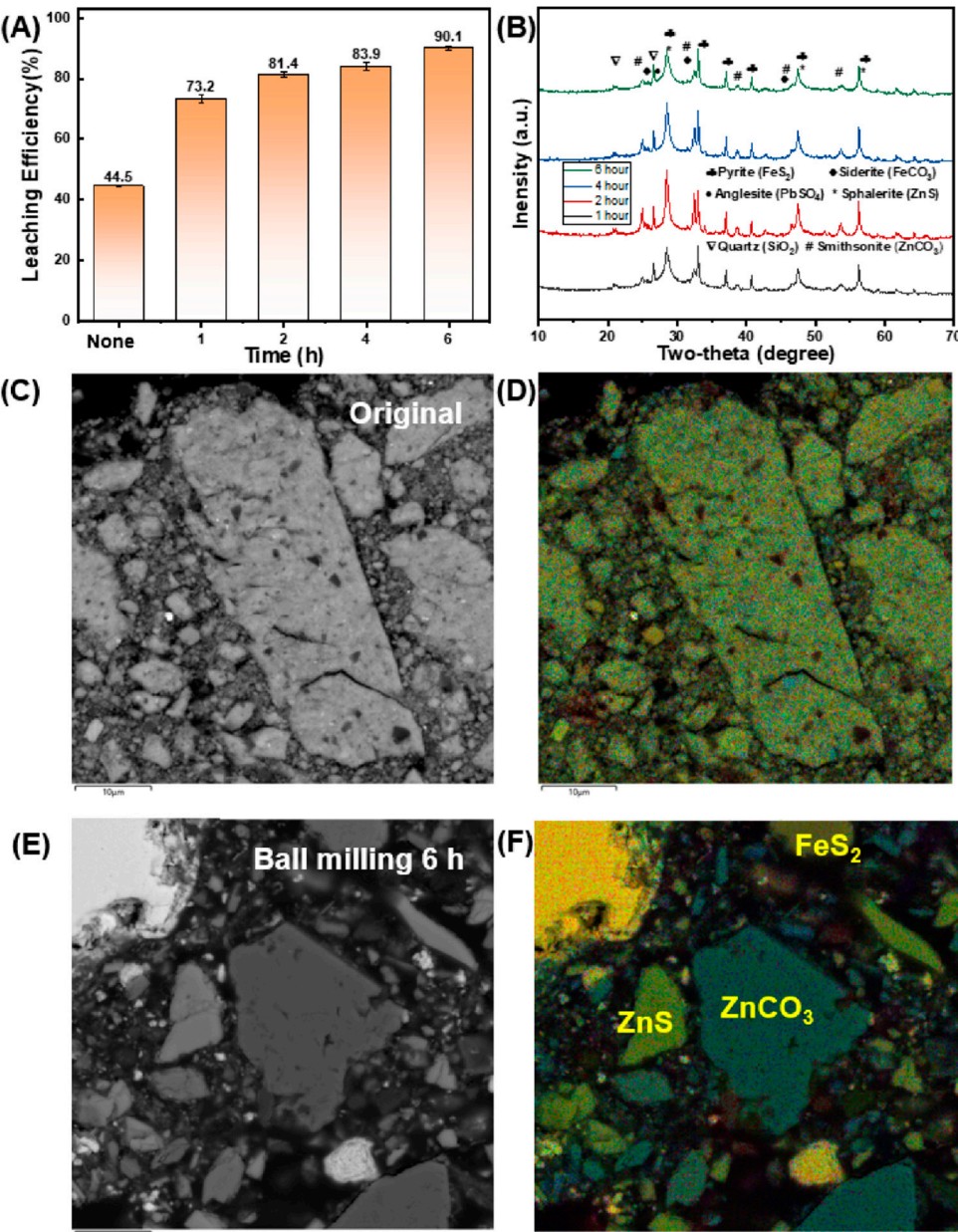

**Figure 5.** (**A**): Effect of rotation time on leaching efficiency [ball/mass ratio: 40:1, rotation speed: 500 rpm]; (**B**) XRD pattern of original CMO and sample treated for 6 h; (**C**) SEM image of original CMO; (**D**) BSE image of original CMO; (**E**) SEM image of CMO sample treated for 6 h; (**F**) BSE image of sample treated for 6 h.

To investigate the composition of amorphous phases, attenuated total reflectance Fourier transform infrared spectroscopy (ATR-FTIR) was employed to analyze the composition of CMO after ball-milling treatment. As shown in Figure S8, the common absorbance peaks at 743 cm$^{-1}$, 866 cm$^{-1}$, and 1405 cm$^{-1}$ were attributed to the typical model of $v4$ $(CO_3)^{2-}$, $v2$ $(CO_3)^{2-}$ and $v3$ $(CO_3)^{2-}$ for $ZnCO_3$ [33]. It is worth noting that absorbance peaks at 778 cm$^{-1}$ and 796 cm$^{-1}$ were observed in CMO with mechanical activation (Figure S8). These peaks were ascribed to the vibration of octahedral zinc correlated with the -OH group [34,35]. Therefore, it was speculated that the main composition of the amorphous phases was the unstable Zn-bearing species (octahedral Zn-OH structure). The octahedral Zn-OH structure was less stable than crystalline ZnS [35]. Thus, the solubility of the amorphous phase might be greater than crystalline ZnS in an organic acid solution. The

increase in leaching reactivity was probably due to the increase in the amorphous phase, which contained unstable Zn-bearing species (octahedral Zn-OH structure).

### 3.4.3. Effect of Rotation Time

The effect of rotation time on Zn leaching efficiency was shown in Figure 5A. It can be seen that as the rotation time increased from 1 to 6 h, the Zn leaching efficiency increased from 73.2% to 90.1%. The Zn leaching efficiency of 90.1% was comparable with recent works (Table S2), while the reaction environment was much milder. This result implied that a higher Zn leaching efficiency could be obtained at a longer time of mechanical activation.

The effect of rotation time on the phase composition and degree of amorphization CMO were studied. The XRD pattern in Figure 5B showed that the typical peaks of ZnS were broadened. Meanwhile, the typical peak of $ZnCO_3$ disappeared gradually with rotation time increasing from 1 to 6 h. This phenomenon indicated that Zn-bearing minerals were transformed into amorphous phases. As revealed in Figure S9, it could be seen that the degree of amorphization increased from 47.1% to 54.1% when the rotation time increased from 1 to 6 h. The signal of unstable octahedral Zn-OH structure was also observed in ATR-FTIR (Figure S10), which implied that the increasing amorphous region was related to the unstable Zn-OH structure in CMO. As discussed above, Zn leaching efficiency was elevated when the proportion of amorphous region increased by ball-milling treatment.

In addition to the composition change in CMO, the changes in morphology and size could further influence the leaching reactivity. In the original CMO, Zn-bearing minerals were mixed with gangue minerals in a bulk rectangle shape, of which the size increased to 40~50 μm long (Figure 5C,D). The ball-milling treatment broke the mixing structure into irregular small particles with sizes in the range of 5~10 μm (Figure 5E). EDS mapping of the particle section demonstrated that single mineral particles (e.g., smithsonite, sphalerite, and pyrite) were liberated from mixed ores (Figure 5F). Therefore, the increased exposure to single Zn-bearing minerals increased the mineral–organic acid interaction, leading to a higher leaching reactivity [36].

### 4. Conclusions

In this study, a combined method of organic acid leaching with mechanical activation was explored for Zn recovery from carbonate-type Pb-Zn mixed ore (CMO). Lactic acid showed the highest selectivity of Zn among the tested organic acids, with the highest leaching efficiency of 38.8%. The optimum reaction conditions were identified as the lactic acid concentration of 1.15 mol/L, L/S ratio of 20, leaching time of 75 min, and temperature of 75 °C. To further increase the separation efficiency, ball-balling treatment was used to activate CMO before organic acid leaching. Under the optimum condition, the Zn leaching efficiency of activated CMO increased from 44.5% to 90.1%. It was found that the degree of the amorphous region in CMO increased after mechanical activation. The main composition of these amorphous phases was unstable Zn-bearing species, including ZnS and $ZnCO_3$. These unstable species showed a higher solubility than the crystalline regions. Moreover, small Zn-bearing particles were liberated from bulk minerals, which increased the exposure of Zn-bearing species to the leaching agent. Thus, the Zn leaching reactivity was elevated after mechanical activation. The combined method could obtain Zn recovery comparable with previous literature results. However, the leaching reagent was much lower, given that the Zn concentration of CMO was higher than in previous reports. This work is believed to contribute to the development of green hydrometallurgy strategies, due to the lower acidity and toxicity of organic acid, as well as the enhanced recovery efficiency combined with mechanical activation.

**Supplementary Materials:** The following supporting information can be downloaded at: https://www.mdpi.com/article/10.3390/met13061021/s1, Table S1: Parameter levels and coded values used in the experimental design; Table S2: Brief Summary of Zn recover from low-graded Zn ores; Table S3: Experiment design; Table S4: Fit Summary; Table S5: ANOVA for Quadratic model; Table S6: ANOVA for Revised Quadratic model; Table S7: pH before and after leaching

process; Table S8: Chemical descriptors of organic acids; Table S9: Optimum conditions by RSM results; Table S10: BET surface area of CMO at different rotation speed; Table S11: BET surface area of CMO at different ball/mass ratio; Figure S1: Inorganic acid leaching at different pH; Figure S2: Organic Acid Leaching efficiency of various metal; Figure S3: Lactic acid leaching at different pH [Lactic acid (1 M), Liquid/Solid ratio: 10 mL/1 g; Time: 1 h; Temperature: 60 °C]; Figure S4: (a) The normal probability plot of externally studentized residuals of the predictable Zn leaching efficiency; (b) the predicted and actual response slope of Zn leaching efficiency; Figure S5: The 3D plot of response surface analysis: (a) L/S ratio and lactic concentration; (b) temperature and time; Figure S6: XRD pattern of leaching residue; Figure S7: Degree of amorphization at different ball/mass ratio; Figure S8: ATR-FTIR spectra of CMO at different ball/mass ratio; Figure S9: Degree of amorphization at different rotation time; Figure S10: ATR-FTIR spectra of CMO at different time. References [37–45] are cited in the supplementary materials.

**Author Contributions:** Investigation, H.D. and X.P.; Methodology, H.D., F.C. and Q.G.; writing— original draft preparation, H.D.; writing—review and editing, H.D. and C.T.; supervision, C.T. and Z.L.; project administration, C.T. and Z.L. All authors have read and agreed to the published version of the manuscript.

**Funding:** This work was carried out with financial support from South China University of Technology. The research was funded by following three fundings: the Natural Science Foundation of Hunan Province of China (No. 2021JC0001); the National Natural Science Foundation of China (No. 22222612); and the National Key Research and Development Program of China (No. 2022YFC3901104).

**Data Availability Statement:** Not applicable.

**Acknowledgments:** Thanks for the training of Central South University and for the guidance and help of the teachers of the Chinese National Engineering Research Center for Control & Treatment of Heavy Metal Pollution.

**Conflicts of Interest:** The authors declare no conflict of interest.

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
