# Peer review of "Enhanced Recovery of Zn from Carbonate-Type Mixed Oxidized Ore (CMO) by Combining Organic Acid Leaching with Mechanical Activation"

_metals, doi:10.3390/met13061021_

Round 1

Reviewer 1 Report

This is a useful study on zinc extraction from ores by organic acid leaching, being more environmentally friendly than that using inorganic acids. Three leaching efficiency parameters relating to the ore preparation were ball to sample mass ratio, rotation rate and time of ball milling. Six parameters of leaching efficiency relating to very leaching procedure were considered:  stirring rate, pH, temperature, acid concentration, time and liquid to solid ratio, last four of which were used to derive statistical leaching efficiency model. Thus it is recommendable to publish the manuscript. Before that, some missing data should be added, and a series of text modifications should be executed in order to improve the readability, as listed below.

P1, L15. To emphasize the limits of the study, all organic acids used should be mentioned in Abstract (also in final paragraph of Introduction section).  Suggestion to begin the sentence in Line 15: Among organic acids applied (lactic, malonic, citric, amber, acetic and tartaric one) the lactic acid was selected for its….

P2, L 35 ..  one of the alter resources..:  vague meaning (perhaps – one of the alternative resources)?

P2, L 73-74  Please insert the details of ball milling leading to given effectiveness, since there is a possibility that more intensive ball milling may additionally improve the leaching efficiency, as Figs 4 and 5 indicate.

P2, L86: ..chemical composition of CMO analyzes by..,  correct to read: ..chemical composition of CMO was analyzed by..

P5, L 173-185. Here the authors should mention the complex compound formation as a factor of Zn extraction (see for instance https://doi.org/10.1016/j.ctta.2021.100024)

General:  Nothing was said in the manuscript about the concentrations of accompanying metals (Pb, Fe, Ca) in the solutions after leaching procedure, nor about the expected troubles to obtain pure zinc from these solutions? A corresponding paragraph or new section should be added.

Table S3, column Leaching efficiency of Zn: What justifies six significant figures in this column?. I suggest that standard measurement accuracy allows maximum three significant figures.

Table S9. The dimensions of quantities in the title row are missing. Check also the number of significant figures in each column.  

The linguistic corrections are suggested within the Comments and Suggestions for Authors

Reviewer 2 Report

Review

Manuscript Number: metals-2398055

Title:  Enhanced Recovery of Zn from carbonate-type mixed oxidized ore (CMO) by combining organic acid leaching with mechanical activation

In this paper, the authors have examined method for Zn recovery from carbonate-type Pb-Zn mixed ore (CMO), by combining organic acid leaching with mechanical activation. It was found that lactic acid had the highest leaching efficiency among tested organic acids and the optimal conditions were: lactic acid concentration of 1.15 mol/L, L/S ratio of 20, leaching time of 75 min, and temperature of 75oC. The leaching efficiency was 90.1%.

There are some questions and remarks to be answered:

1.    The authors should perform a proof reading of the text (some mistakes, typos, etc.).

2.    There is a lack of graphical abstract.

3.    There is no information on the applied reagents (chemical purity, producer).

4.    There is a lack of information, concerning the applied method of error analysis, method accuracy and reproducibility.

5.    In the Conclusions, the authors should explain why the proposed method is better than the existing ones, give its advantages and compare the obtained results with previous literature results.

6.    What practical application do the authors envisage for the proposed method?

The authors should perform a proof reading of the text (some mistakes, typos, etc.).

Round 2

Reviewer 2 Report

The authors referred to all comments accordingly.